# Study on the Modification of Confining Rock for Protecting Coal Roadways against Impact Loads from a Roof Stratum

**Haiyang Yi** [1]**, Zhenhua Ouyang** [1,]*****, Xinxin Zhou** [1]**, Zhengsheng Li** [1]**, Jianqiang Chen** [2]**, Kang Li** [2] **and Kunlun Liu** [2]

1  School of Safety Supervision, North China Institute of Science and Technology, Langfang 065201, China;
   haiyangyi@ncist.edu.cn (H.Y.); zxx08522580@163.com (X.Z.); lizhengsheng0512@163.com (Z.L.)
2  Shenhua Xinjiang Energy Co., Ltd., Urumchi 830027, China; jianqiangchen_sh@163.com (J.C.);
   kangli_sh@163.com (K.L.); kunlunliu_sh@163.com (K.L.)
*  Correspondence: oyzhua@163.com

**Abstract:** Promoting the ability of anti-bursting of the confining rock of a coal roadway is of significant importance to the safe production of a coal mine. In particular, in deep-buried coal mines, highly frequent rock burst occurs due to large earth pressure and complex geological conditions, which needs serious improvement. This paper investigated a type of confining rock modified method, which can modify the physical properties of the surrounding rock and form a crack region and a reinforced region by blasting and grouting reinforcement. Based on a set of physical model experiments and numerical modeling, the results of a comparative analysis between a normal roadway and the modified roadway in the static stress redistribution, dynamic stress, damage evolution, and energy dissipation suggest that the modified confining rock is capable of protecting the coal roadway against rock burst from roof stratum, obviously reducing and transferring the concentered static–dynamic stress out of the cracked region, dissipating the dynamic energy by plastic damage in the cracked region, and keeping the integrity of the reinforced region. In addition, the velocity of the dynamic stress vibration wave at the surface of the modified coal roadway is obviously reduced, which is beneficial for decreasing the movement of cracked rock blocks and protecting the lives and goods in the coal roadway.

**Keywords:** rock burst; confining rock modification; energy dissipation; coal roadway; stress transfer

## 1. Introduction

Rock burst during coal mining is a typical dynamic disaster that threatens the safety production seriously in coal mining country [1–4]. Despite a serious of engineering techniques that have been announced to be successful in preventing rock burst during coal mining, rock burst occurs occasionally and induces serious damage in coal roadways or even death; in addition, coal mines in China with high potential rock burst are increasingly emerging due to the increasing mining depth [5]. Therefore, capable approaches that protect the coal roadway against rock burst are urgently desired for coal mining in China.

Roadway support, including wood, arch, metal, bolting, and hydraulic supports, are commonly and widely used to control the deformation and stabilize the confining rock of roadways. With the growth of mining depth, bolt–anchor systems associated with grids by steel belts, arched sheds or hydraulic support are primary applied against the large displacement or broken zone induced by large in situ stress, while such traditional roadway supports were normally designed in the view of static pressure [6–9]. In recent years, the increasing number of accidents in coal roadways induced by rock burst brought awareness to researchers that the ability of coal roadways in impact resistance is of serious importance to the safety of deep coal mining. Thus, various techniques were developed to dissipate the dynamic energy of rock burst. These techniques can be mainly categorized into three types [10]: namely, energy dissipation by damping rock bolts [11–14], energy absorption by high-power hydraulic devices [15–17], and retractable steel sheds [18–20]. The damping

rock bolts release a certain degree of strain energy by sliding friction or damper [21–23], the hydraulic devices act as dampers that release dynamic stress from confining rocks, thereby stabilizing the coal roadway [24], and the retractable steel sheds bear the load from confining rocks and dissipate the dynamic stress by its retractable structure [25,26]. Sometimes, these techniques were combined to improve the resisting ability of the coal roadway for rock burst [27,28].

Although the effectiveness of the above techniques for protecting the coal roadway against rock burst has been proofed in many engineering cases, several serious accidents occurred in recent years, such as the rock burst in Qianqiu coal mine [29], Yuejin coal mine [30], Sunjiawan coal mine [31], Longyun coal mine [32], etc., which indicate that the current ability of coal roadways in coal mines is still not enough to protect against the strong rock burst, exhibiting large and sudden deformation of coal roadways or even crushed rock pieces rushed into the coal roadway, consequently causing damage to equipment or even death [29]. In addition, mining the deep coal resource in China is inevitable due to the characters of coal resource distribution and the large amount of coal assumption in China. The high stress concertation, strong disturbance, and complex lithological structure would increase the intensity of rock burst [32]. Therefore, a new approach to promoting the resistant ability of a coal roadway against rock burst is becoming an increasingly important issue for coal mines.

Kaiser and Cai [33] outlined the design principal of a rock burst supporting system, including rock-burst prevention, yielding support, addressing the weakest link, using an integrated system that is simple and cost effective, and being adaptable, and the basic functions of an effective supporting system, namely, reinforcing the rock mass, retaining broken rock, and holding fractured blocks. In the light of these design principals and functions, this paper studied a new type of coal roadway, an in situ confining rock-modified coal roadway, which strengthens the surrounding rock, releases the nearby concentered stress, and dissipates the kinetic energy by the cracked rock zone. Detailed information on this type of modified coal roadway is introduced in the next section. Similar roadway structures were investigated by Wang et al. [17] and Gao et al. [18,34], in terms of the mechanism from the view of theoretical analysis and laboratory experiments. The difference between the studied roadway in this paper and these similar roadways is the structural formation of a reinforced region or cracked region. Specifically, the previously studied roadways still require an inner supporting system to stabilize the broken rocks or an artificial seismic isolated structure to protect the roadway against rock burst; in our case, the roadway is designed as a formation with in situ modification of confining rock, which is desired to reduce the cost of construction of a coal roadway. Anyway, previous studies on similar roadways illustrate that it is feasible to protect the roadway against rock burst with a reinforced and a weakened zone, while the capability of the confining rock modified coal roadway in this paper needs a primary study and discussion.

Motivated by the above purpose, this paper launched a physical experiment based on the geological conditions of a real coal mine, and a corresponding full-scaled numerical model was computed as well, targeting the capability of the purposed in situ modified coal roadway associated with analysis of the stress redistribution, dynamic response, and energy dissipation. Additional discussion on the feasibility of engineering implementation of the studied coal roadway and its economic benefits is carried out as well in this paper, and conclusions are summarized in the end.

## 2. Conception of Modification in the Confining Rock of a Coal Roadway

As illustrated in Figure 1, the modification of a coal roadway contains a cracked region and a reinforced region. The cracked region can be formated by multi-times blasting with a small quantity of charge, hydrofracturing technology or supercritical carbon dioxide blasting, depending on the geological and engineering conditions. The reinforced region is designed to be strengthened by grouting. These two regions were concept formation, which is still in further study, in terms of cracking methods, cracking and reinforcement

ranges, cracking degrees, construction sequences, etc. The feasibility of modifying a coal roadway as shown in Figure 1 is discussed in detail in Section 6. The modification of the confining rock in a coal roadway is designed with three targets: namely, transferring the concentrated stress far away from the tunnel side to a relative further distance, dissipating the kinetic energy out of the coal roadway, and stabilizing the coal roadway.

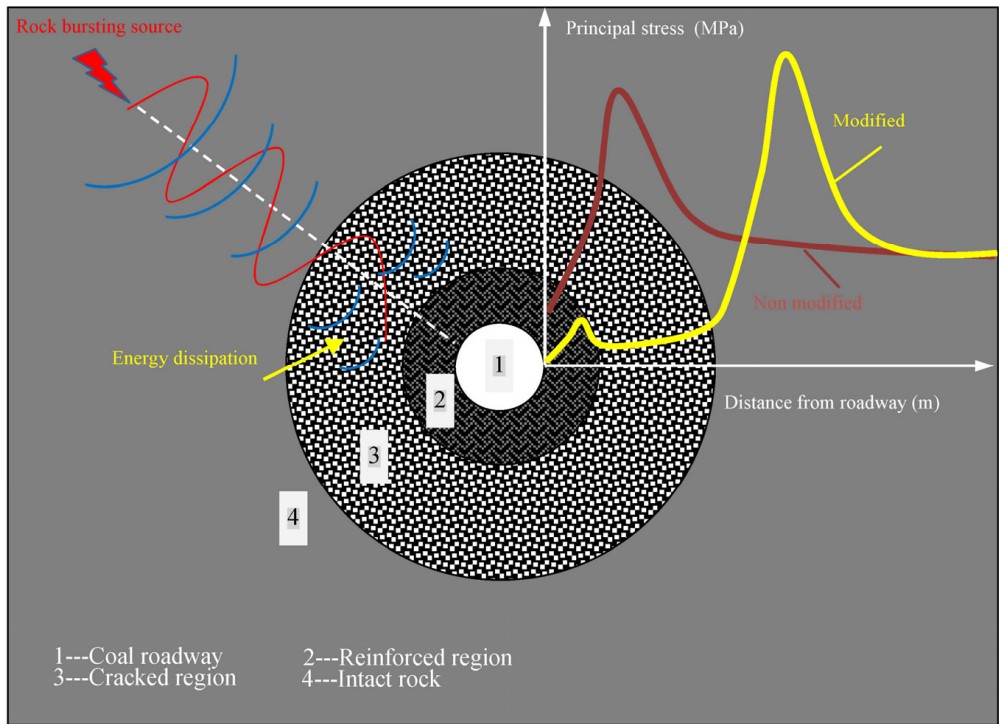

**Figure 1.** Generalized model of a modified coal roadway.

Specifically, a stress concentrated area exists near the tunnel side (the junction of plastic and elastic areas) [35,36], as the curves plotted in Figure 1 show; if the stress concentrated area was cracked, as denoted by Zhang et al. [37], the primary stress would be transferred to the outer boundary of the cracked region. For the purpose of kinetic energy dissipation, as suggested by Yan et al. [38], Cho et al. [39], and Tang et al. [40], damaged rock is able to increase its energy dissipation density. In the case of a modified coal roadway, as illustrated by the schematic diagram in Figure 1, most of the kinetic energy from the rock burst source can be adsorbed by the cracked region, thereby protecting the coal roadway. Additionally, grouting is a common method to enhance the strength and integrity of rocks around coal roadway [41]; therefore, a grout-reinforced region in the modified coal roadway as displayed in Figure 1 is able to promote the supporting ability of the coal roadway.

In order to investigate the feasibility of the above targets achieved by a modified coal roadway, physical experiments and numerical modeling were conducted based on the real geological conditions of a real coal mine, the Kuangou coal mine. A detailed engineering background of the studied engineering case is introduced in the following section.

## 3. Engineering Background

The geological conditions of the Kuangou coal mine were taken as an engineering example. The Kuangou coal mine is located at Queergou of Hutubi County in Urumuqi City of Xinjiang Province. The area of the coal seam in Kuangou coal mine is about 21 km$^2$, and its designed annual production is about 1.2 million tones. The B$_2$ coal seam is the primary excavating source, whose average thickness and dip angle are 5.6 m and 14°, respectively. Geological investigation by borehole suggests that (please see Table 1) the average depth of the B$_2$ coal is about 480 m, and three potential hard and thick seams,

namely the coarse sandstone (8 m and 7.6 m in average thickness) at 33.9 m and 65.6 m above the $B_2$ coal seam are the high potential sources of roof bursting to the $B_2$ coal seam.

**Table 1.** Lithological characteristics of the Kuangou coal mine investigated by borehole.

| No. | Hatch Pattern | Lithology | Average Depth/m | Average Thickness/m | Cumulated Distance from $B_2$ Coal Seam/m |
|---|---|---|---|---|---|
| 1 | | Coarse sandstone | 366.1 | 14.8 | 121.4 |
| 2 | | Siltstone | 380.9 | 9 | 106.6 |
| 3 | | Fine sandstone | 389.9 | 26 | 97.6 |
| 4 | | Sandy mudstone | 415.9 | 6 | 71.6 |
| 5 | | Coarse sandstone | 421.9 | 7.6 | 65.6 |
| 6 | | $B_4^2$ coal seam | 429.5 | 1.5 | 58 |
| 7 | | Fine sandstone | 431 | 9.6 | 56.5 |
| 8 | | $B_4^1$ coal seam | 440.6 | 4 | 46.9 |
| 9 | | Fine sandstone | 444.6 | 9 | 42.9 |
| 10 | | Coarse sandstone | 453.6 | <u>8</u> | 33.9 |
| 11 | | $B_3$ coal seam | 461.6 | 1.8 | 25.9 |
| 12 | | Siltstone | 463.4 | 4.8 | 24.1 |
| 13 | | Fine sandstone | 468.2 | 19.3 | 19.3 |
| 14 | | $B_2$ coal seam | 480 | 11.8 | 0 |
| 15 | | Sandy mudstone | 484 | 4 | −11.8 |
| 16 | | Fine sandstone | 505 | 21 | −15.8 |
| 17 | | $B_1$ coal seam | 510.6 | 5.6 | −36.8 |
| 18 | | Sandy mudstone | 525.6 | 15 | −42.4 |
| 19 | | Coarse sandstone | 531.5 | 5.9 | −57.4 |

The $B_2$ coal and floor have a weak burst tendency, while the roof has a strong burst tendency, which are the inner factors for rock burst. Actually, one of the most serious bursting events (about $9.7 \times 10^6$ J) recorded in 2018 occurred in the coarse sandstone that

is 33.9 m above $B_2$ coal seam, which caused a 20 cm hump at the tunnel floor and 30 cm subsidence at the tunnel roof.

## 4. Methodology

A physical model experiment and numerical modeling were involved in studying the energy dissipation of in situ modification of a coal roadway under the roof bursting. Taking the real engineering conditions of the Kuangou coal mine in Xinjiang Province as an example, the numerical model and physical model were constructed based on the real lithological structure. Detailed information on the numerical model setup and physical model experiment are introduced in the following subsections.

### 4.1. Physical Model Experiment

### 4.1.1. Similarity Relationship

The physical quantities of the physical model were determined according to the theory of similarity in geometry, mechanics, and dynamics. The geometric length, density, and elastic modulus were selected as the fundamental dimensions and defined with the ratios of the prototype to model as 100:1, 1.5:1, and 150:1, respectively. As both the prototype and scaled model were in the same acceleration field, the ratio of acceleration is 1:1. In the light of a dimensional similitude principle, alternative physical quantities were calculated as listed in Table 2, in which the symbol *c* represents the similarity ratio, and the subscripts *p* and *m* denotes the prototype and physical model, respectively. *l*, *ρ*, *E*, *δ*, *σ*, *c*, *φ*, *ε*, *v*, *V*, *α* and *F* represent the length, density, elastic modulus, displacement, stress, cohesion, friction angle, strain, Poisson's ratio, velocity, acceleration, and force, respectively. The physical and mechanical parameters of rocks are listed in Table 3, which were provided by the Kuangou coal mine and determined by laboratory tests.

**Table 2.** Similarity proportions of the physical quantities for physical model.

| Quantities | Similitude Relations | Ratios | Quantities | Similitude Relations | Ratios |
|---|---|---|---|---|---|
| Length | $C_l = l_p/l_m$ | 100 | Strain | $C_\varepsilon = \varepsilon_p/\varepsilon_m = 1$ | 1 |
| Density | $C_\rho = \rho_p/\rho_m$ | 1.5 | Poisson's ratio | $C_v = v_p/v_m = 1$ | 1 |
| Elastic modulus | $C_E = E_p/E_m = C_\rho C_g C_l$ | 150 | Velocity | $C_V = V_p/V_m = C_E^{0.5} C_\rho^{-0.5}$ | 10 |
| Displacement | $C_\delta = \delta_p/\delta_m = C_l$ | 100 | Time | $C_T = T_p/T_m = C_L C_V^{-1}$ | 10 |
| Stress | $C_\sigma = \sigma_p/\sigma_m = C_E$ | 150 | Acceleration | $C_\alpha = \alpha_p/\alpha_m = C_g$ | 1 |
| Cohesion | $C_c = c_p/c_m = C_E$ | 150 | Force | $C_F = F_p/F_m = C_\rho C_g C_l^3$ | $1.5 \times 10^6$ |
| Friction angle | $C_\varphi = \varphi_p/\varphi_m = 1$ | 1 | Energy | $C_e = F_e/F_e = C_F C_l$ | $1.5 \times 10^8$ |

**Table 3.** The physical and mechanical parameters of rocks.

| Rock Type | Density (kg·m$^{-3}$) | Young's Modulus (MPa) | Poisson's Ratio (-) | Cohesion (MPa) | Friction Angle (⁰) | Tensile Strength (MPa) |
|---|---|---|---|---|---|---|
| Coarse sandstone | 2530 | 5990 | 0.18 | 6.57 | 39.2 | 5.21 |
| Fine sandstone | 2580 | 4090 | 0.2 | 5.42 | 37 | 4.2 |
| Siltstone | 2570 | 2250 | 0.2 | 4.43 | 37.4 | 3.28 |
| Sandy mudstone | 2510 | 3425 | 0.21 | 3.16 | 36 | 2.75 |
| Coal | 1335 | 1530 | 0.25 | 2.21 | 30.3 | 1.64 |
| Reinforce region | 1335 | 1193 | 0.22 | 2.84 | 34.3 | 2.46 |
| Cracked region | 1600 | 918 | 0.28 | 1.32 | 20 | 0.6 |

### 4.1.2. Physical Model Setup

Prior to the construction of the physical model, a series of orthogonal tests and consolidation tests were carried out to determine the similar materials of rock layers and modified regions of the coal roadway. Corresponding to the similarity relations of density and Young's modulus of the rock materials (see Table 3) and designed rock regions, as shown in Table 4, the rock layers were simulated by mixtures of river sand, slacked lime,

plaster, and water. The reinforced region was modeled by the same similar materials of the coal seam with a higher proportion of 3 wt % plaster and less river sand to increase its elastic modulus (about 30% increasement related to cracked region), and that of the cracked region contains 14 wt % plastic particles to modeled cracks and decreases its elastic modulus (approximately 60% degradation related to intact coal).

**Table 4.** Similar materials of rock layers and modified regions of a coal roadway.

| Modeled Rock | Consolidated Thickness (cm) | Weight (g) | | | | | |
|---|---|---|---|---|---|---|---|
| | | River Sand | Slacked Lime | Plaster | Water | Kieselguhr | Plastic Particle |
| Siltstone | 11 | 51.98 | 0 | 5.78 | 2.89 | 0 | 0 |
| Fine sandstone | 24 | 112 | 4.67 | 14 | 6.3 | 0 | 0 |
| Coal | 12 | 56.7 | 1.09 | 6.3 | 3.15 | 0 | 0 |
| Sandy mudstone | 4 | 18.9 | 2.68 | 2.1 | 1.05 | 0 | 0 |
| Coarse sandstone | 29 | 133.22 | 12.72 | 19 | 7.61 | 0 | 0 |
| Reinforced region | - | 42.5 | 2.64 | 32.6 | 28 | 3 | 0 |
| Cracked region | - | 56.7 | 1.09 | 8.3 | 6.15 | 0 | 14 |

The model was compacted layer-by-layer (see Figure 2a). When it came to the model of the roadway, for the regular one, a polyvinyl chloride (PVC) tube with an outer diameter of about 40 mm was inserted into the model layers, which was pulled out to simulate the regular roadway. For the modified roadway, as shown in Figure 2b, the model of reinforced and cracked regions was pre-formed in a cylinder made by a piece of aluminum sheet; after the model of the modified region was installed in the corresponding layer (see Figure 2c) and the next upper layer was compacted, the cylinder was taken out. After all the model layers were finished, the overburden wight of ground layers (about 300 m) was modeled by iron blocks (about 0.28 m in height and 21.5 kPa; see Figure 2d). During the construction of the physical model, four small dimensional pressure sensors were placed in the model at the points (N1, N2, M1, M2) shown in Figure 2e, which were arranged along the radial direction of the cylinder. M1 and M2 were located at the outer surface of the reinforced and cracked regions, while N1 and N2 were the same height as that of M1 and M2. The applied sensors are self-developed by the Institute of Engineering Mechanics, CEA, the minimum diameter of the sensor is 10.5 mm, the thickness is 5 mm, the natural response frequency is 100–300 kHz, the range of stress measurement is 0–1.0 MPa, and the applicable temperature is −40 to + 125 °C. It can resist strong and weak electromagnetic, high centrifugal force, strong impact, and interference.

Finally, the physical model was constructed in a two-dimensional format with a height of 1.15 m, a width of 2 m, and a thickness of 0.2 m. As plotted in Figure 2f, the coal roadways were built in the scaled $B_2$ coal seam, which is 0.5 m away from the model bottom and sides, and 0.65 m away from the model top. According to the lithological structure in Table 1, based on the $B_2$ coal seam, the upward layers to the coarse sandstone (65.6 m from the $B_2$ coal seam) and the downward layers to another coarse sandstone (−57.4 m from the $B_2$ coal seam) were contained in the physical model. As a whole, corresponding to the thickness of each rock layer and the similarity ratios in Tables 1 and 2, the scaled model contains 15 layers and two scaled coal roadways. The inner diameters of the coal roadways in Figure 2f are about 40 mm, and the left scaled roadway is a regular one, while the right roadway is a modified one with a reinforced region (80 mm in diameter) and a cracked region (160 mm in diameter). The center of the roadway is 0.65 m from the top of the model and 0.5 m from the bottom and side of the model. The distance between the centers of the two roadways is 1 m.

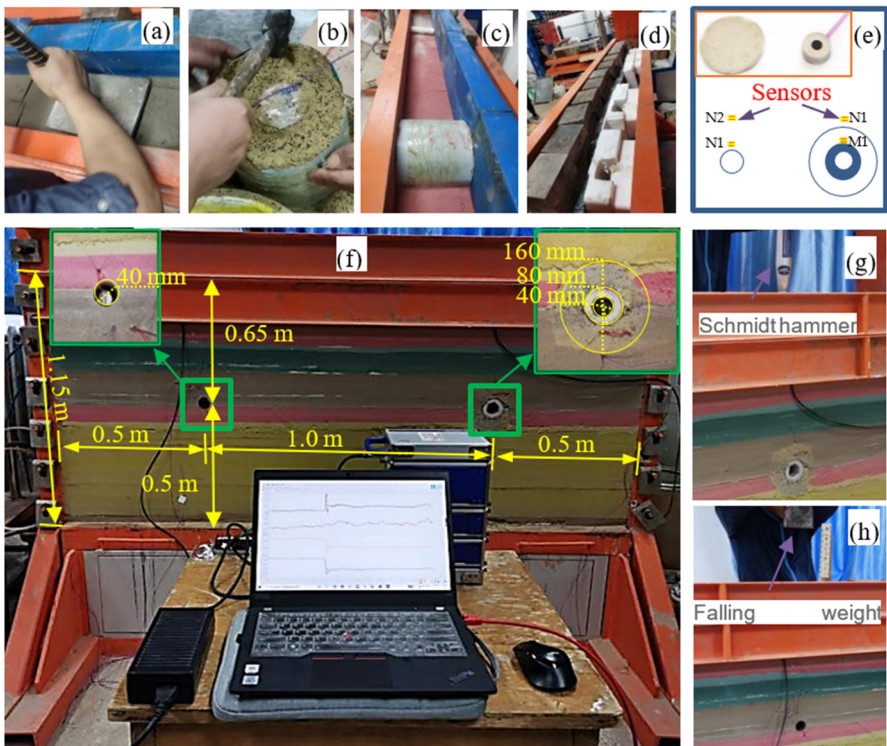

**Figure 2.** Field of the physical model experiment. (**a**–**d**) illustrate the building processes of the model included layer-by-layer compaction, construction of the modified roadway, insertion of the roadway models and additional bob-weight loading, respectively, (**e**–**h**) show the arrangement of the pressure sensors, the experiment field, Schmidt hammer loading and falling weight impact loading, respectively.

### 4.1.3. Impact Loading

Two types of load methods were applied to the physical model, as displayed in Figure 2g,h: namely, Schmidt hammer and falling weight impact. The Schmidt hammer has a standard impact energy output of about 2.207 J (about $3.3 \times 10^8$ J in the prototype), the falling weight is about 55 N, and the falling height is around 0.2 m; thus, the energy of the falling weight impact is about 11 J (about $16.5 \times 10^8$ J).

### 4.2. Numerical Model Setup

#### 4.2.1. Software Introduction

Abaqus is a finite element software for engineering simulation, which can be applied for linear and nonlinear analysis. Especially, the explicit analysis module in abaqus is suitable for analyzing transient dynamic events such as impact and explosion. This paper mainly investigates the wave dissipation effect and the law of energy dissipation at the modified roadway under impact dynamic load, so abaqus can be used to simulate this process well.

#### 4.2.2. Model Characters

Corresponding to the lithological structure of the Kuangou coal mine as listed in Table 1 and the physical model, a numerical model was built as displayed in Figure 3, which is a full-scale model with a length of 200 m, a width of 2 m, and a height of 130 m. In light of the simulated relations of length of the prototype to the model, the inner diameter of the coal roadway is 4 m, the outer diameter of the reinforced region is 8 m, and that of the cracked region is 16 m. The rock layers and locations of the coal roadways are the same as that of the physical model. Structured grid division is adopted globally, with a total of 200,000 units. The unit type is hexahedron. The grid in some areas of the roadway

and surrounding rock is relatively dense. By adjusting the parameters of the reinforcement zone and crushing zone, the effect of reinforcement and crushing can be achieved. The Mohr–Coulomb model was used for calculation.

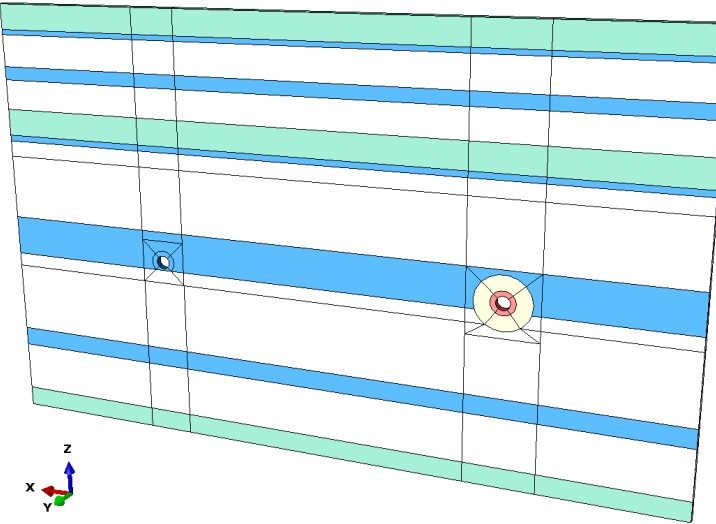

**Figure 3.** Numerical model of coal roadway and rock layers.

### 4.2.3. Parameters Assignment

The parameters in Table 3 were utilized in the numerical modeling. The global damping was applied in the numerical model. According to the method of determining the parameters of Rayleigh damping coefficients by frequency analysis, the Rayleigh damping coefficients of the studied model were calculated as $\alpha = 1.41912$, $\beta = 0.0017$.

### 4.2.4. Boundary Conditions

In the numerical modeling of a dynamic model, the methods of using infinite boundary elements and artificial boundaries are commonly applied to solve the issues of boundary dynamic refection [42,43]. In our case, the method of artificial boundary proposed by Su et al. [44] was applied to avoid the scattering waves reflected from the truncated geological bodies. Specifically, in the static computation, the left and right boundaries of the numerical model were fixed in the displacement along the X-direction, the front and back boundaries were fixed in the displacement of the Y-direction, and the bottom boundary was fixed in the Z-direction. The top surface applied an additional pressure of 6.9 MPa (corresponding to the 46.2 kPa in the physical model). Alternatively, elastic–viscosity elements were set at the nodes of all the outer surfaces with the coefficients [45] of spring modulus $k_b = G/2r_b$ and viscous damper $c_b = \rho c_s$, in which $G$ and $\rho$ are the shear modulus and mass density of the modeled rock layers, respectively, $r_b$ is the radius of the truncated formation away from the impact source, and $c_s$ is the propagation velocity of the impact wave.

### 4.2.5. Simulated Impact Loads

In order to verify the physical model, the impact loads in the physical model were simulated in the numerical model. According to the parameters of a standard N-type Schmidt hammer, the peak velocity of the piston rode is about 3.366 m/s, and the contact velocity is about 1.98 m/s in the case of weight falling load. Corresponding to the similitude relation of velocity, the amplitude of the Schmidt hammer wave velocities and weight falling load in the prototype are 33.66 m/s and 19.8 m/s, respectively. In the numerical modeling, the simulated impact loads of the Schmidt hammer and weight falling were added at a mesh node and a surface (equivalent area of the iron block).

According to the statistic results of micro seismic signals by He et al. [24], the majority of frequency induced by a large energy event of rock burst is around 5 Hz.

## 5. Results and Analysis

### 5.1. Comparison of the Results of Numerical Model and Physical Model

The results of numerical modeling (pink curves) and amplified data (green scatters) of the physical experiment under the impact load of a Schmidt hammer are plotted in Figure 4. The data of the physical experiment were enlarged by 150 times according to the similarity ratio of stress as listed in Table 2. When the impact load came from the top surface above the normal coal roadway, please see Figure 4a, the point N2 had the largest vertical dynamic stress, the second one the that of point N1, and the peak values of M1 and M2 were close to each other. When the impact load was added to the top surface of the modified roadway, as displayed in Figure 4b, the maximum vertical dynamic stress occurred at M2, while that of point M1 placed second.

As a whole, the numerical modeling results fit well with the physical experiment in terms of the peak value of the first dynamic stress and the range of the progressive stress value, which verified that the numerical modeling reproduced well the dynamic response of the confining rock of both the normal and modified roadway.

### 5.2. Static Stress Redistribution

Stress is one of the key factors that induces rock burst, large deformation, or even broken rock in the confining rock of a coal roadway. As displayed in Figure 5a, the majority of the concentrated stress near the roadway is significantly transferred out of the cracked region in the modified coal roadway, and the peak value of stress of a modified roadway is about 1 MPa smaller than that of a normal roadway. For a normal roadway, the maximum principal stress is concentrated in the rock nearby tunnel side at about 1 m in depth (see Figure 5b), while for a modified coal roadway, the areas of stress concentration appeared out of the cracked region (see Figure 5c), which is larger than that of a normal roadway. This means the degree of elastic energy concentration is reduced by the modification of confining rock of a roadway; as a result, the potential of rock burst can be eliminated to some extent by the decreased peak stress value and enlarged bearing areas of elastic energy.

In addition, Figure 5b,c suggest that the tensile stress (positive value) in the vault and inverted arch of a modified coal roadway are far smaller than that of a normal roadway, and the maximum tensile stress around the modified roadway is located near the bottom of the cracked region. This means the potential of bending or breaks in the vault and the inverted arch of a modified roadway would be eliminated significantly, thereby protecting the integrity of the confining rock.

### 5.3. Dynamic Response

In order to analyze the effects of a modified coal roadway on dissipating the dynamic energy from the impact load to a coal roadway and stabilizing the confining rock of a coal roadway, comparative analysis on a normal coal roadway and a modified coal roadway, in terms of the dynamic stress, damage evolution, and energy dissipation were conducted as the following subjections.

#### 5.3.1. Dynamic Stress

The curves of calculated and measured dynamic stress versus time around a normal coal roadway and a modified coal road are plotted in Figure 6a,b, respectively. In addition, the contour pictures of dynamic stress and total stress at various times (corresponding to time points A and B in the curves) are also displayed in Figure 6 as well. The calculated curves of dynamic stress at point N1 and M1 fit well the data (which were amplified by 150 times according to the similarity ratio) of physical experiment, which suggest that the numerical modeling is capable of reproducing the real impacting process.

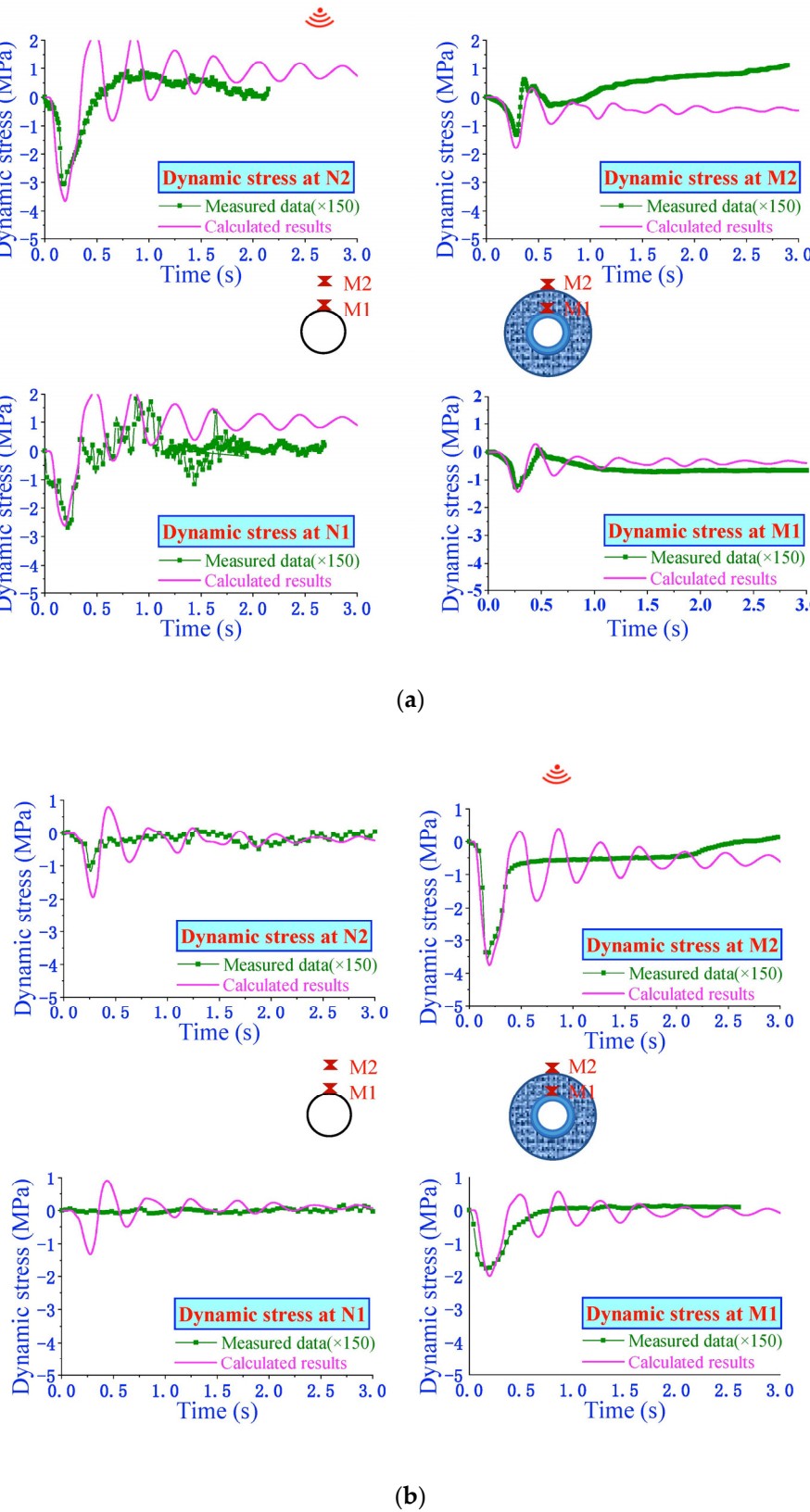

**Figure 4.** Experiment and numerical results of vertical dynamic stress induced by impact load of Schmidt hammer from the top right above (**a**) a normal coal roadway and (**b**) a modified coal roadway.

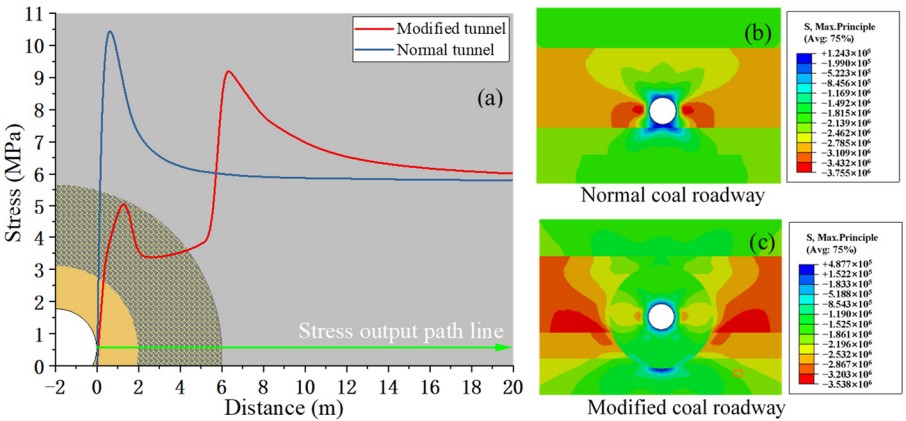

**Figure 5.** Maximum principal stress distribution (**a**) along a path line away from the tunnel roadway, (**b**) contour of stress around a normal coal roadway and (**c**) a modified roadway.

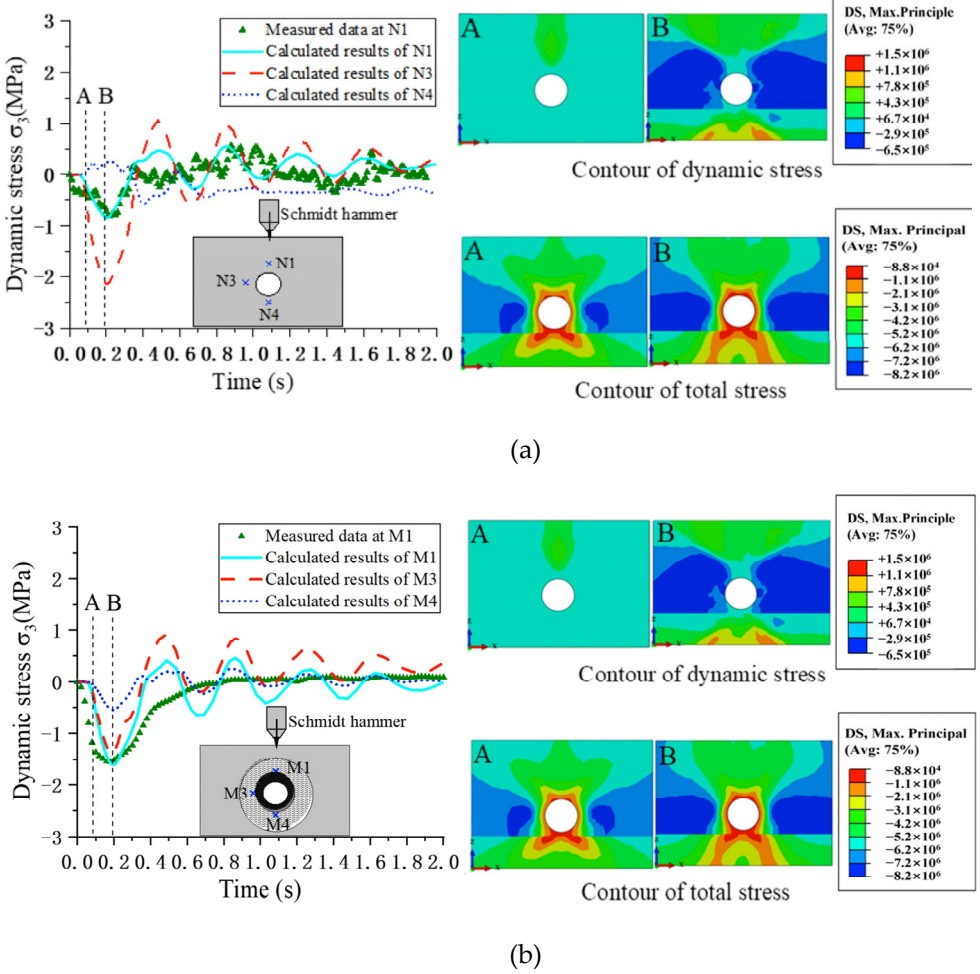

**Figure 6.** Dynamic stress and contour of stress distribution of (**a**) a normal coal roadway and (**b**) a modified coal roadway under the impact load of a Schmidt hammer.

Obviously, when dynamic waves propagated to the normal coal roadway, the point of N3 (side wall) had the largest compressive dynamic stress, about $-2.2$ MPa at around 0.18 s. While in the case of a modified roadway, the largest compressive stress aroused at the side wall (point M3) was approximately $-1.4$ MPa, which was close to the data of point M1. Correspondingly, as the contour pictures of dynamic stress in Figure 6 show, the side walls of a normal coal roadway were subjected to the largest compressive principal

stress at about 0.18 s, while for the confining rock of a modified rock, small areas of the side walls had the compressive principal stress, while the majority of the compressive principal dynamic stress occurred at the intact rock out of the cracked region and small areas at the vault and invert arch in the cracked region. As a whole, comparing the total stress distribution around a normal roadway and a modified coal roadway (see the contour pictures in Figure 6) shows that the impact load from the roof layers induces a large area of the side wall of a normal coal roadway, which would damage the integrity of the roadway, while cracked and reinforced regions in the modified coal road help transfer the static-dynamic stress out of the cracked region, thereby protecting the integrity of the coal roadway.

### 5.3.2. Damage Evolution

The evolution of plastic damage that occurred around the normal and modified coal roadway is displayed in Figures 7 and 8, respectively.

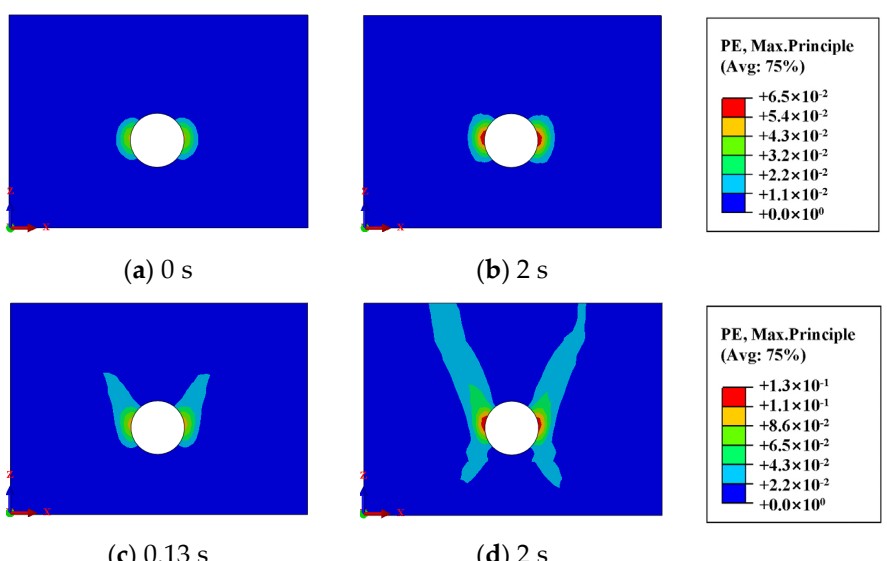

**Figure 7.** Evolution of plastic damage of the normal coal roadway under the impact load of (**a**,**b**) the Schmidt hammer at 0 s and 2 s and the impact load of (**c**,**d**) the weight fall at 0.13 s and 2 s, respectively.

For a normal coal roadway, as shown in Figure 7a,b, a smaller impact load (Schmidt hammer) enlarged the plastic damage area slightly at the side wall, while the larger impact load (weight fall) led to cracking through the roof and floor (see Figure 7c,d); serious cracks through the whole coal layer formed a high potential caving zone tendency.

For the modified coal roadway, when a smaller impact load was loaded (please see Figure 8a,b), small areas above the roadway in the cracked region were damaged. However, as shown in Figure 8c,d, the majority of the crack region, some area of the intact rock above the roadway, and a small part of the reinforced region at the side wall experienced plastic damage. A comparison between Figures 7b and 8b and between Figures 7d and 8d indicate that the cracked region in the modified coal roadway has the ability of dissipating dynamic energy by releasing the elastic energy in the cracked region or even in the outer intact rock.

It is notable that the reinforced region (see Figure 8b,d) was protected well despite the small damage that occurred at the side wall during the large impact loading process, which means that the modification method in this paper is able to keep the integrity of the nearby confining rock of the coal roadway, thereby protecting the safety of lives and goods in the roadway.

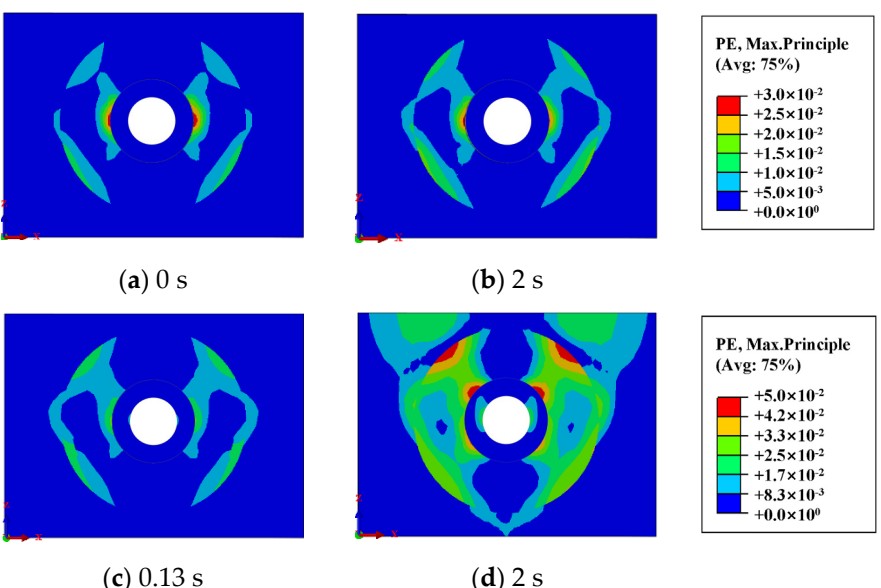

(**a**) 0 s          (**b**) 2 s

(**c**) 0.13 s          (**d**) 2 s

**Figure 8.** Evolution of plastic damage of the modified coal roadway under the impact load of (**a**,**b**) the Schmidt hammer at 0 s and 2 s and the impact load of (**c**,**d**) the weight fall at 0.13 s and 2 s, respectively.

### 5.3.3. Energy Dissipation

The velocity curves of the points at the outer surfaces of the reinforced and cracked regions (M1 and M2, respectively) and those of the points at N1 and N2 (the same height as that of M1 and M2) above the normal coal roadway are illustrated in Figure 9. In order to investigate the kinetic energy dissipation along the impact source to the coal roadway, here, we analyze the negative peak values of the curves in Figure 9; the negative value means the direction of the velocity is along the propagation direction of the dynamic stress.

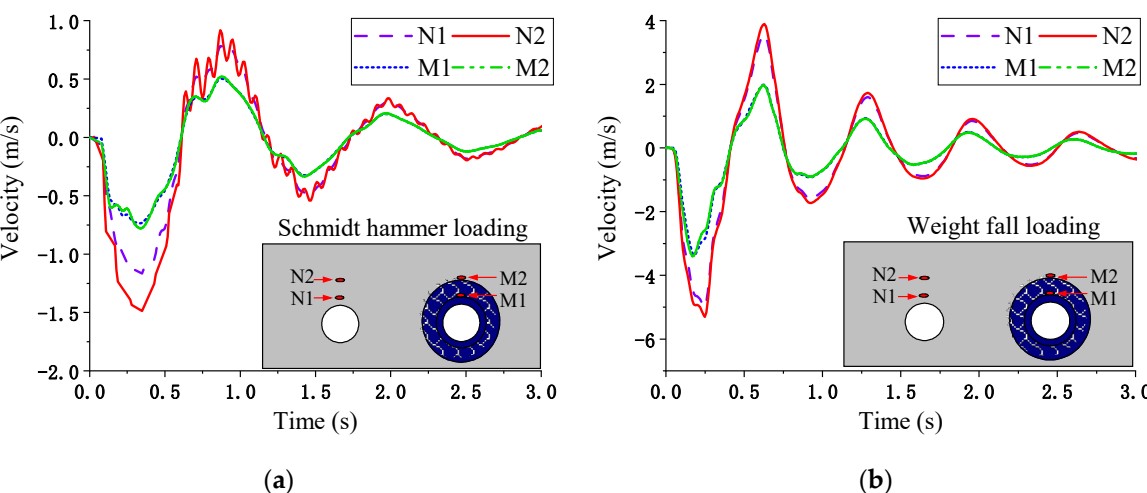

(**a**)          (**b**)

**Figure 9.** Calculated velocity curves above the coal roadway under (**a**) Schmidt hammer loading and (**b**) weight fall loading.

In the case of Schmidt hammer loading, as shown in Figure 9a and Table 5, the maximum velocity at point N2 above the normal coal roadway is about 1.5 m/s, while when the dynamic wave arrived at point N1, the velocity was reduced to about 1.2 m/s. This was caused by the dynamic scattering near a hole. Interestingly, the peak values of the velocities at the points of M1 and M2 were close to each other; such a phenomenon was mainly caused by the cracked regions which led to dynamic scattering before point M2 on one hand, and the reinforced region let the confining rock of the roadway move unitedly.

As a whole, the maximum velocity of the rock above the modified coal roadway was approximately 50% to 62.5% smaller than that above the normal coal roadway. The velocity of the above points of observation under weight fall loading, as plotted in Figure 9b, showed the similar tendency to that under Schmidt hammer loading. The difference is the velocity of point N1, which was close to that of point N2; this phenomenon is caused by the large cracking damage as plotted in Figure 7d, which led to the united movement in the rock area that contained points N1 and N2. Despite this difference, Figure 9b tells us that the modified coal roadway is still capable of reducing the velocity of rocks nearby the roadway, exhibiting the maximum velocity at points M1 and M2, which is about 64% of that at points N1 and N2.

**Table 5.** The velocity above the coal roadway under Schmidt hammer loading and weight fall loading.

| Velocity (m/s) | N1 | N2 | M1 | M2 |
|---|---|---|---|---|
| Schmidt hammer loading | 1.2 | 1.5 | 0.75 | 0.76 |
| Weight fall loading | 5 | 5.4 | 3.44 | 3.45 |

The output of elastic strain energy density along the propagation of dynamic load from the top surface to the normal and modified roadway are plotted in Figure 10a,b, respectively.

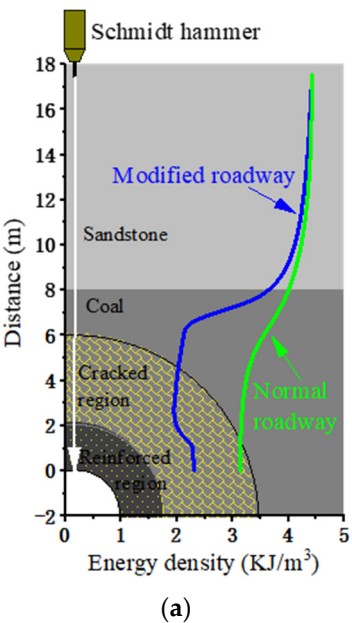 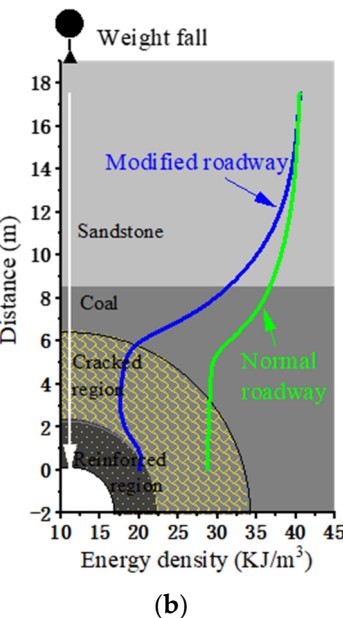

(**a**)      (**b**)

**Figure 10.** Energy density along the propagation path of impact load by (**a**) a Schmidt hammer and (**b**) a falling weight above the normal roadway and modified roadway.

Figure 10 and Table 6 suggest that the kinetic energy density decreased along the propagation path of the impact load, especially through the interface of the sandstone and coal layers due to wave scattering. Obviously, in the case of modified coal roadway, the decrease in energy density was more dramatic than that above a normal coal roadway. Specifically, under the impact load of a Schmidt hammer, please see Figure 10a, the energy density was about 4.2 kJ/m$^3$ in the sandstone, which was reduced to about 3.1 kJ/m$^3$ at the inner top surface of the normal coal roadway. For a modified coal roadway, the energy density was decreased to about 2 kJ/m$^3$ in the cracked region initially and increased slightly to about 2.3 kJ/m$^3$ in the reinforced region. Despite this increase, the energy density at the vault of a modified coal roadway was about 74% that of a normal coal roadway. As shown in Figure 10b, the evolution of energy density under the impact load of a falling weight

was similar to that in Figure 10a, and the percentage of the energy density at the vault of a modified coal roadway compared to that of a normal coal roadway is about 71%.

**Table 6.** Energy density along the propagation path of impact load by a Schmidt hammer and a falling weight above the normal roadway and modified roadway.

| Energy Density (KJ/m³) | Normal Roadway | | Modified Roadway | |
| --- | --- | --- | --- | --- |
| | Sandstone | Top Surface | Crack Region | Reinforced Region |
| Schmidt hammer loading | 4.2 | 3.1 | 2 | 2.3 |
| Weight fall loading | 40 | 28.8 | 17.5 | 20 |

As a whole, Figures 9 and 10 convince us that the modified coal roadway reduces the dynamic energy of the impact load propagated in the confining rock significantly.

## 6. Discussion

The above analysis on the stress redistribution, dynamic response of confining rock, and energy dissipation suggest that the modification of a coal roadway by cracked and reinforced regions is beneficial for eliminating stress concentration nearby the roadway, dissipating kinetic energy and improving the integrity of the confining rock. Actually, this kind of structure for the kinetic energy dissipation has been applied in subway tunnels, as announced by the literature of [46–48], where buffer layers and concrete linings were set as the damping and reinforcement components, respectively.

Both the study of this paper and a previous investigation on subway tunnels indicate that the modification of the confining rock of a coal roadway is feasible to protect the roadway against the impact load from rock roofs. However, this paper only proposed a primary idea, and the numerical simulation and similarity model experiment mainly illustrated this effect of the modified roadway from a qualitative analysis angle, the specific parameters about the crack region and reinforced region, grouting materials, etc., and the economic costs of such a modification whether it is cheaper than a common supporting roadway or not, which need to be studied further.

## 7. Conclusions

The present paper carried out a set of physical model experiments and numerical modeling to investigate the capability of the confining rock modification, namely constructing a cracked region and a reinforced one, for a coal roadway to protect itself against rock burst from roof stratum. The results of numerical modeling match well the measured data in the physical modeling, which suggest that the numerical model is able to reproduce the real condition of the impacting processes of the coal roadway. Through comparative analysis between a normal and a modified coal roadway, in terms of the static stress redistribution after roadway construction and the dynamic response of the confining rock under two types of impact loads, the following conclusions can be drawn:

1.  After the implementation of the confining rock modification of a coal roadway, the stress concentration at the side wall of the roadway will be transferred out of the cracked region and the peak rock pressure will be reduced, thereby eliminating rock burst at the side wall of the coal roadway and protecting the integrality of the confining rock of the coal roadway as well.
2.  Under the impact load from the roof stratum, the maximum dynamic stress occurred mainly in the intact rock out of the cracked region, and the maximum static–dynamic stress is still distributed out of the cracked region. In addition, the plastic damage in the cracked region under impact load helps absorb the dynamic energy and protect the integrality of the confining rock of a coal roadway.

3.  The modification of a coal roadway in confining rock decreases the velocity and dynamic energy density significantly at the vault of a coal roadway; therefore, it is capable of reducing the movement of the cracked blocks from the surface of a coal roadway and protecting the staffs or goods in the coal roadway.

**Author Contributions:** Conceptualization, Z.O.; Data curation, X.Z.; Investigation, Z.L.; Methodology, H.Y.; Validation, J.C. and K.L. (Kang Li); Visualization, K.L. (Kunlun Liu); Writing—original draft, H.Y. All authors have read and agreed to the published version of the manuscript.

**Funding:** The study of this paper was founded by the National Natural Science Foundation of China (Grant No.51974125, 52004090), the National Natural Science Foundation of Hebei Province (Grant No. E2020508002, E2020508025), the Fundamental Research Funds for the Central Universities (No.3142018001, 3142019019) and scientific and technological research project founded by Education department of Hebei Province (Grant No. QN2019320).

**Data Availability Statement:** The data supporting the study are included in this paper.

**Acknowledgments:** Acknowledgments are also given to the support of Tianchi Hundred People Program of Xinjiang Uygur Autonomous Region (2019(39)) herein.

**Conflicts of Interest:** Authors Jianqiang Chen, Kang Li and Kunlun Liu are employed by the Shenhua Xinjiang Energy Co. The remaining authors declare that the research was conducted in the absence of any commercial or financial relationships that could be construed as a potential conflict of interest.

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
