# Peer review of "Study on the Modification of Confining Rock for Protecting Coal Roadways against Impact Loads from a Roof Stratum"

_minerals, doi:10.3390/min11121331_

Round 1

Reviewer 1 Report

Comments on minerals-1437107

  1. The list of references absolutely unbalanced, almost all the references are from Chinese scholars. The Western, Russian, Poland, and some other countries studies where the rock burst disasters were intensively investigated are cited very poor and even ignored, the citation style must be essentially improved
  2. The English style is poor and many grammatical problems can be found. For example, at lines 12-13, what asks for ability, is that rock burst? At line 20, the velocity of what, is that “dynamic stress vibration wave”? At line 66, verb expressions such as “avoid rockburst” should be changed to nouns
  3. In the abstract, the authors point out the research method of confining rock modification. However, there is no mention of which aspects of confining rock are modified, strength or materials? Some kind of explanations must be added
  4. Section 4.1.2, page 4, lines 178-179. The authors mentioned small dimensional pressure sensors, please supplement the text description of the sensor, including sensor installation method, transmission mode, number of layout and specific location. Is it evenly arranged along the axial direction of the cylinder? Is the method of comparing the obtained pressure data through the physical model experiment with the stress value obtained by numerical simulation scientific
  5. Page 6, lines 185-188. What were the criteria for selecting the dimensions of distance between the normal roadway and the modified roadway, the distance away from the model top, model bottom and sides
  6. Section 4.1.3, page 6, lines 200-202. The energy of Schmidt hammer and falling weight impact are 3.3×108J and 16.5×108J respectively. However, the actual energy of a large bursting event is 9.7×106J showed at line 133. Why did the author choose the energy two orders of magnitude larger than the actual energy as the data of the physical model experiment
  7. Section 4.2.4, page 8, line 243. The authors took 5Hz as the frequency of large energy event but there is no explanation to this, nor information about mine operations was provided. Maybe the frequency is related to seismic wave parameters which changes with different large energy event and different coal mines.
  8. The chapter number is wrong at lines 203, 265, 287 and 293.
  9. “ground stress” should be changed to “in-situ stress”.
  10. The word “protentional” at lines 31, 275, and 281 is wrong which should be “potential”.
  11. The word “will” at line 261 should be changed to “well”.
  12. The author used repeated expressions many times, like “convince us……” or “which convince us……”, the style can be improved.
  13. For Figure 2, the letter number in the upper right corner of the picture is obstructed.
  14. For Figure 10, the unit of energy density should be kJ/m3, not kJ.
  15. At line 242, the statistic results of micro seismic signals were from He et al, please quote relevant literature.

Author Response

Dear reviewer, thanks greatly for your great comments and suggestions on our paper manuscripts, which improved the quality of our paper to a large extent. We accept all your opinions and the corrected points have been marked by red color or track formation.

Reviewer 2 Report

  1. Section 4.2.1 does not indicate the software used by the numerical simulation model, the calculation model used by the model, and the grid division of the judgment model?
  2. Section 4.1 of line 203 is marked incorrectly and should be section 4.2.
  3. Partial occlusion marked in Figure 2, please modify it.
  4. It can be seen from FIG. 5 (b) and c that the planes between different layers of the model are not connected, and the model is not connected as a whole. There are obvious dislocation and discontinuity in the stress cloud map. The same problem appears in other cloud images.
  5. Section 4.2 The model has a high degree of fit with the real situation, but the idea is not open. It is only verification, which cannot fully reflect the value of the model. However, it is recommended to supplement the research on "weak zones". How thick should the zones be to make them work well?
  6. In Section 4.2.3, the boundary setting only considers geo-stress caused by gravity, not tectonic geo-stress, and there is no setting of relevant horizontal stress content.
  7. In the upper right corner of Fig.6 (a), the content of selected cloud image options is wrong.
  8. The expressions of relevant energy values in Section 5.3.3 are all in the text, so it is difficult to make an intuitive comparison. Supplementary diagrams are needed for illustration.
  9. In the discussion part, the content given is consistent with the effect not given. We all know that this kind of structure is conducive to reducing the stress concentration level, but there is basically no specific content. Need to give a deeper, thought-provoking discussion.

Author Response

Thanks greatly for your great comments and suggestions on our paper manuscripts, which improved the quality of our paper to a large extent. We accept all your opinions and the corrected points have been marked by red color or track formation.

Reviewer 3 Report

Although, nowadays, the transition to green energy is based, based on unconventional energy sources, the energy formed by coal mining remains a central pillar in energy production.

The authors of the article successfully investigate the anti-bursting ability of the confining rock of coal roadways. The evaluation from the point of view of the voltage distribution, of the dynamic voltage and of the evolution of the deterioration and energy dissipation model led to the modeling of a modified roadway.

Please note the following observations: in figure 1 units of measurement for the axes of the graph must be introduced; for a better understanding please explain in more detail the points N1, N2, M1, M2.

Please specify if you have taken into account the fact that, on a normal scale, various defects may occur in the rock / sand / clay layers that constitute stress concentrators.

Author Response

(The authors gave the same response as above.)

Reviewer 4 Report

The authors discuss an interesting topic. However, there several thinks that must be amend

  • The whole text requires a revision due to there are a few mistakes
  • The abstract should be improved, highlighting the novelty of the research done
  • Introduction should be improved, including some of the principal references of the field. The authors MUST go to the source. For instance
    • lines 34-39 cite a reference about support and static pressure that is not the fundamental reference of that topic, there are a lot of previous research on the field that should be cited instead of the one chosen from 2019.
    • Lines 43-51: cites are included in bulk and there are also many previous and relevant research that stablished the fundamentals of what is explained.
    • Sentence from lines 89 to 91 are irrelevant, the authors should consider delate them.
  • Section 2. The general explanation in lines 106-108 should also include the proper citations. Despite the authors currently cited could have very good papers published on the topic, they are not the only ones and, as I already said, it is necessary to include the fundamental research on the topic.
  • Section 3. Please rephrase lines 123-133 as the explanation is not clear enough.
  • What software was used for the numerical model? Ore information about the model used and its characteristics should be included

Author Response

(The authors gave the same response as above.)

Round 2

Reviewer 1 Report

All questions have been dealed with.

Reviewer 4 Report

The authora have changed all the points. No additional comments.